# A Combination of β-Aescin and Newly Synthesized Alkylamidobetaines as Modern Components Eradicating the Biofilms of Multidrug-Resistant Clinical Strains of *Candida glabrata*

**DOI:** 10.3390/ijms25052541

**Published:** 2024-02-22

**Authors:** Emil Paluch, Olga Bortkiewicz, Jarosław Widelski, Anna Duda-Madej, Michał Gleńsk, Urszula Nawrot, Łukasz Lamch, Daria Długowska, Beata Sobieszczańska, Kazimiera A. Wilk

**Affiliations:** 1Department of Microbiology, Faculty of Medicine, Wroclaw Medical University, 50-376 Wroclaw, Poland; olga.bortkiewicz@umw.edu.pl (O.B.); anna.duda-madej@umed.wroc.pl (A.D.-M.); beata.sobieszczanska@umw.edu.pl (B.S.); 2Department of Pharmacognosy with Medicinal Plants Garden, Lublin Medical University, 20-093 Lublin, Poland; jaroslaw.widelski@umlub.pl; 3Department of Pharmacognosy and Herbal Medicines, Wroclaw Medical University, 50-556 Wroclaw, Poland; michal.glensk@umw.edu.pl; 4Department of Pharmaceutical Microbiology and Parasitology, Wroclaw Medical University, 50-556 Wroclaw, Poland; urszula.nawrot@umw.edu.pl; 5Department of Engineering and Technology of Chemical Processes, Wroclaw University of Science and Technology, 50-370 Wroclaw, Poland; lukasz.lamch@pwr.edu.pl (Ł.L.); 252671@student.pwr.edu.pl (D.D.); kazimiera.wilk@pwr.edu.pl (K.A.W.)

**Keywords:** β-aescin, newly synthesized alkylamidobetaines, surface-active compounds, surfactant, multidrug resistant, *Candida glabrata*, biofilm, biological activity

## Abstract

The current trend in microbiological research aimed at limiting the development of biofilms of multidrug-resistant microorganisms is increasingly towards the search for possible synergistic effects between various compounds. This work presents a combination of a naturally occurring compound, β-aescin, newly synthesized alkylamidobetaines (AABs) with a general structure—C_n_TMDAB, and antifungal drugs. The research we conducted consists of several stages. The first stage concerns determining biological activity (antifungal) against selected multidrug-resistant strains of *Candida glabrata* (*C. glabrata*) with the highest ability to form biofilms. The second stage of this study determined the activity of β-aescin combinations with antifungal compounds and alkylamidobetaines. In the next stage of this study, the ability to eradicate a biofilm on the polystyrene surface of the combination of β-aescin with alkylamidobetaines was examined. It has been shown that the combination of β-aescin and alkylamidobetaine can firmly remove biofilms and reduce their viability. The last stage of this research was to determine the safety regarding the cytotoxicity of both β-aescin and alkylamidobetaines. Previous studies on the fibroblast cell line have shown that C9 alkylamidobetaine can be safely used as a component of anti-biofilm compounds. This research increases the level of knowledge about the practical possibilities of using anti-biofilm compounds in combined therapies against *C. glabrata*.

## 1. Introduction

In addition to their obvious benefits, the widespread and decades-long use of antibiotics and antifungal drugs has resulted in several negative phenomena (e.g., widespread disruption of intestinal flora and overuse of antibiotics in animal husbandry) and the great challenge facing modern medicine, namely the resistance of microorganisms to antimicrobial drugs. Antimicrobial resistance has been assessed as a hazard, where prevention, a defective treatment of the ever-expanding number of pathogens, becomes difficult or impossible. One of the main factors that can initiate the development of a *Candida* infection and prolong its duration is the formation of a biofilm [1,2].

A biofilm is an organized microbial community embedded in a protective extracellular matrix; it is immersed in this friendly environment on biotic and abiotic surfaces [2]. The biofilm formation process by different types of pathogens causes the possibility of developing drug resistance and deep tissue and systemic mycoses [3].

The formation of biofilm structures by *Candida* spp. has been noticed on numerous and various types of surfaces, including living surfaces (e.g., internal organs, mucous membranes, or blood vessels) as well as non-living surfaces (e.g., in the indoor environment of the hospital, including the floors, walls, fittings, fixtures, and furniture) and medical equipment that have direct contact with patients’ bodies (e.g., stents, implants, endotracheal tubes, and different types of catheters) [2,4,5]. 

Among the over 150 species of the genus *Candida*, only a few are considered pathogenic and responsible for human or animal diseases. However, it is an excellent challenge for medicine and the healthcare system because opportunistic yeasts are responsible for 90% of cases of systemic mycoses, causing approximately 1.6 million deaths each year [6,7]. *Candida albicans* (*C. albicans*) remains the primary human fungal pathogen [8] and, in fact, the most frequently isolated from healthy and infected organisms (80% isolates from all forms representing human candidiasis [9]). 

However, over the last few years, an increasing number of infections caused by non-albicans representatives of the *Candida* genus, such as *C. glabrata*, has been observed worldwide [8,10,11].

*C. glabrata* (*Nakaseomyces glabrata* in new terminology) has been recognized as a type of human commensal yeast that is part of the normal microbiota with the ability to colonize mucosal surfaces (the oral cavity as well as the gastrointestinal and genital tracts) that are usually harmless or only cause mild infections [8,12].

*C. glabrata*, under certain favorable conditions, acts as an opportunistic pathogen, being the cause of both superficial (mucocutaneous infection) and systemic infections. Moreover, life-threatening invasive candidemia concerns critically ill patients, often with an immunodeficiency [13,14]. This is reflected in the statistics according to which *C. glabrata* is one of the leading causes of invasive candidiasis in Europe (up to 20% of all cases) [15]. Clinical isolates of *C. glabrata* show intrinsic as well as acquired antifungal resistance to commonly used drugs (especially to azoles [16]), which may lead to treatment failure. This is why infections caused by *C. glabrata* are often characterized by higher morbidity and mortality rates than those caused by *C. albicans* [8,17]. The limitation of antifungal therapeutics to only several classes of drugs, namely polyenes, echinocandins, triazole derivatives, and low efficiency, is the reason for not fulfilling expectations [18]. 

This alarming trend is mainly the consequence of the ability to form biofilms, which is one of the main virulence factors [19,20], confirmed by the presence of biofilms in more than 80% of human infections [21,22]. Yeasts embedded in biofilms increase their resistance to antifungal substances [19,20] and present up to 1000-fold higher resistance levels than their planktonic forms [18,23].

The primary strategy to overcome the urgent and crucial problem of fungal resistance relies on searching for new substances with anticandidal activity and combination therapy or combining existing antifungal drugs with non-antifungal agents [12]. The phytochemicals from many plant species exert inhibitory activities against many fungal pathogens when used alone or in combination with traditionally used antifungal drugs [24].

β-aescin (β-AE) is one of the cardinal phytochemicals of the natural mixtures of triterpene saponins (called aescin) occurring in *Aesculus hippocastanum* L. (*Hippocastanaceae*) seeds. β-AE is commonly used for its beneficial role in therapy, considering its anti-edematous, anti-inflammatory, and vasoprotective effects [12,25]. 

β-AE belongs to the great family of natural surfactants called saponins [26]. Natural compounds from the saponins group consist of an aglycone (with hydrophobic properties), which is connected with polar (hydrophilic) oligosaccharide chains. This characteristic amphiphilic nature of saponin compounds makes them applicable in, among others, industry, pharmacy, and medicine [27,28,29]. Moreover, β-AE shows antifungal activity against planktonic *Candida* cells and biofilms as an anti-biofilm agent that can prevent biofilm formation and the destruction of mature ones [12]. 

Another group of surfactants (other than the natural surfactant represented by β-AE) used in our studies comprised betaine derivatives—AABs. Betaines belongs to zwitterionic surfactants commonly used in personal and home care products [30]. The key functional groups in the structure of betaines may indicate the quaternized nitrogen and the carboxylic group [30].

Betaines and their derivatives (sulfobetaine, AABs) are essential classes of surfactants due to their distinct and unique properties, e.g., dispersing and emulsifying ability, high skin compatibility, and low level of irritation, as well as good chemical stability in the presence of other surfactants [31,32,33]. It should be noted that betaine-type surfactants may be mixed with all other classes of surfactants, i.e., cationic, anionic, and non-ionic ones, to enhance their application properties. 

These facts explain the great interest in studies on betaines and their derivatives. Moreover, betaines, which contain in their structure a hydrophobic chain of 8 to 20 carbon atoms, are surface-active agents that are often added to other classes of surfactants to improve their performance, mildness, foaming ability, viscosity, and other desirable properties [34,35].

In the present contribution, we designed and synthesized new [(3-alkanoyilomethyoamine)propyl] dimethylammonium acetates (C_n_TMDAB), comprising a chemodegradable tertiary amide moiety between their hydrophilic headgroup and hydrophobic chain, characterized by their balanced stability and ability to undergo biodegradation. 

The main goal of the presented studies was to evaluate the combination of β-AE with newly synthesized AABs as an efficient agent against the biofilms of multidrug-resistant clinical strains of *C. glabrata*.

The search for new drugs to treat mycoses is essential for addressing current challenges in antifungal therapy, including resistance, limited options of curing, and the need for improved safety and efficacy. Continued research in this field is critical for advancing medical care and improving the outcomes for the individuals with fungal infections. The research presented below is characterized by its high level of innovation and has a chance for the practical use of anti-biofilm compounds in combined therapies against multidrug-resistant *C. glabrata* strains.

## 2. Results

### 2.1. The Synthesis and Characterization of C_n_TMDAB Surfactants

CnTMDAB surfactants (AABs) were synthesized through the quaternization of amideamine-type derivatives, i.e., *N*-[3-(dimethylamine)propyl]-*N*-methylalkylamides (Figure 1). This reaction was conducted at an elevated temperature (ca. 75–115 °C) in low-molecular-weight alcohol (preferably methanol or ethanol) with a 25% molar excess of quaternizing agent (sodium chloroacetate). To avoid unwanted reactions (e.g., oxidation or oxygen-initiated chain reactions), the mentioned process was preceded by carefully flushing the reaction vessel with inert gas. C_n_TMDAB surfactants, after their crystallization from methanol—ethyl acetate, were obtained as white, hygroscopic solids. 

The chemical structures of C_n_TMDAB surfactants (AABs) were confirmed through ^1^H NMR and ESI-MS, while their purity was assessed through elementary analyses (see the data presented in Table 1). The obtained zwitterionic surfactants were also characterized by their melting points and Krafft temperatures (see Table 1). Determining the melting points for the derivatives with decyl alkyl chains (C9) was impossible; the mentioned compound was an amorphous solid at room temperature. Derivatives with ten or twenty carbon atoms in alkyl chains were even soluble in cold water (Krafft temperatures below 10 °C or 0 °C for C9; for longer alkyl chains, C_n_TMDAB surfactants (AABs) were characterized by Krafft temperatures exceeding 20 °C, so their activity may be explained by their limited aqueous solubility at low temperatures. The analyses showed that the synthesis and purification steps yielded the desired compounds with sharp melting points (with the melting range not exceeding 0 °C). 

### 2.2. Microbiological Investigation

The present research’s primary purpose was to evaluate the potential use of a combination of β-AE and newly synthesized AABs as modern components for eradicating fungal biofilms. The microbiological investigation was performed in three stages, which are described below:I.In the first stage, we aimed to select clinically drug-resistant strains of *C. glabrata* with the most vigorous intensity of biofilm production. Among the eight multidrug-resistant clinical strains of *C. glabrata*, the 2586 and 2853 strains were selected. Moreover, the reference strain *C. glabrata* ATCC 90030 was used.II.Further research evaluated the antimicrobial properties of β-aescin, newly synthesized AABs, and antifungal drugs (including the combination) (through their MIC_90_, MFC, and FICI values). Combinations of the tested compounds and their actions with antifungal medications were considered. An essential point of this research was determining the ability of biofilm eradication of compounds against *C. glabrata* strains. The anti-biofilm assays included qualitative and quantitative analyses of the impact of the combined compounds on fungal biofilms. These measurements were performed using fluorescence microscopy with computational analysis of the obtained pictures.III.The last stage was focused on the measurements of the potential toxicity of β-aescin and AABs. For this purpose, a hemolysis test and MTT assays on Balb/3T3 mouse embryonic fibroblasts were conducted to determine the usefulness of these compounds for medicine.

The detailed results of each experiment are described in the following paragraphs.

#### 2.2.1. Selection of *Candida glabrata* Clinical Strains Based on Their Biofilm Production

The selection of *C. glabrata* clinical strains at the beginning of our research was performed using the colorimetric and fluorescence-based measurements of biofilm production. The obtained results are presented in Figure 2. Among all the tested drug-resistant strains, the *C. glabrata* 2586 and 2853 clinical strains presented the most robust outputs of biofilms.

#### 2.2.2. Evaluation of the MIC and MFC of Antifungal Compounds against Selected Strains

In the case of the new derivatives (Table 2) of AABs, C9H19 exhibited activity for the reference strain of *C. glabrata* and 2586 clinical isolates with MIC = 3 µM/mL, which represents 985.47 µg/mL, and for the 2853 clinical isolate, MIC = 1 µM/mL = 328.49 µg/mL. For the rest of the AABs, for all the tested *C. glabrata* strains, the MIC value was the same and equal to 3 µM/mL, which was equivalent to 1069.62 µg/mL for C_11_H_23_, 1153.8 µg/mL for C_13_H_27_, and 1237.95 µg/mL for C_15_H_31_.

β-AE (Table 3) resulted in the inhibition of all the tested *C. glabrata* strains; the MIC_90_ value was 128 µg/mL for the reference strain (MFC = 256 µg/mL), 126 µg/mL for the 2853 clinical strain, and 256 µg/mL for the 2586 clinical strain (in the case of both clinical strains, MIC = MFC). Concerning the fungistatic drugs, the results of antifungal activity are presented in Table 3. It is worth mentioning that the most vigorous activity was shown against caspofungin (MIC = MFC = 0.06 for the reference strain of *C. glabrata* and MIC = 0.03 µg/mL; MFC = 0.06 µg/mL for both clinical strains) and amphotericin B (MIC = 0.25 µg/mL for all tested strains). 

#### 2.2.3. Evaluation of the Fractional Inhibitory Concentration Index (FICI) of β-AE and Antifungal Drugs

In the next step of our studies, the effect of the combination of β-AE with different antifungal drugs (fluconazole, itraconazole, ketoconazole, voriconazole, posaconazole, caspofungin, 5-fluorocytosine, and amphotericin B) on the growth of the reference strain *C. glabrata* ATCC 90030 and the two selected multidrug-resistant strains *2586* and *2853* was evaluated. The possible synergistic effects for all combinations of β-AE with antifungal drugs for each strain, assessed by the calculation of the FICI of each combination, are presented in Figure 3.

The data presented in Figure 3 showed three different types of interactions between β-AE and antifungal drugs: synergistic, additive, and neutral.

The fractional inhibitory concentration index (FICI) only indicated a synergistic effect for β-AE with fluconazole for the ATCC 90030 strain. An additive effect was observed in the combination of β-AE with VOR and POS (ATCC 90030 strain), β-AE with VOR and AMP (*2586* clinical strain), and β-AE with FLU, KET, and POS. In the case of caspofungin (CAS), which presented the lowest MIC values against all the tested strains, β-AE showed an additive effect. It is worth noting that an antagonism interaction was not found.

#### 2.2.4. Interaction of AAB with β-AE and Selected Antifungal Agents

In the next stage of our studies, an evaluation of these interactions, the newly synthesized AABs, at a constant concentration of 3 µM, with β-AE and selected antifungal drugs used in fungal infections (fluconazole, itraconazole, ketoconazole, voriconazole, posaconazole, caspofungin, 5-fluorocytosine, and amphotericin B) was performed. The results were presented as multiplicity factor (MF) values, which were calculated to determine the interactions between the AABs and antimycotics.

It was observed that for itraconazole, caspofungin, and 5-fluorocytosine with AABs for all of the tested *C. glabrata* strains, the interaction did not occur (MF = 1) (see Table 4). All tested AABs (C9, C11, C13, and C15) increased by two times the level of activity of β-AE for a drug-resistant strain of *C. glabrata*. In contrast, the interaction was not observed for the reference strain ATCC 90030 and the second drug-resistant strain 2852.

In the case of fluconazole-tested AAB, it increased its activity by 16-fold (C9) or 64-fold (C11–C15) for the reference strain ATCC 90030, 8-fold (C9) or 64-fold (C11–C15) for the 2586 drug-resistant strain as well as for the drug-resistant strain 2853, 2-fold, (C11 and C13), or 16-fold (C15), while C9 did not interact with the fungicides. The activity of ketoconazole was increased by 16-fold (C9–C13) or 64-fold (C15) for the reference strain ATCC 90030, and by 8-, 16-, 31- and 64-fold (interactions with C9, C11, C13, and C15, respectively) for the drug-resistant strain 2586. For the drug-resistant strain of *C. glabrata*, only interactions with C13 (MF = 2) and C15 (MF = 4) were observed. 

Voriconazole’s activity was increased by 64-fold by all the tested AABs for the reference strain and 2586 drug-resistant strain of *C. glabrata* (only in the case of the 2586 strain, the MF value for C9 was equal to eight). The tested compounds increased the activity of voriconazole (C11 and C13) twice against the drug-resistant 2853 strain, while C9 did not alter the level of activity, and C15 decreased the level of activity by 128-fold.

The AABs (C11 and C13) increased the activity of amphotericin B by 2-fold against all the tested strains of *C. glabrata*, while C15 increased the activity of amphotericin B by 4-fold. In the case of the C9 derivative at the reference strain ATCC 90030 and drug-resistant strain 2853, MC = 1, and for the 2586 strain, MC = 2. 

The worst interaction (reduced activity) the AABs exerted was with posaconazole. For the drug-resistant strain 2853, the activity was 8-(for C9), 4-(for C11 and C13), and even 256-fold times lower; for the reference strain ATCC 90030, posaconazole in the presence of AABs decreased its activity by two-fold with C9 and C15 and increased its activity by 8-fold with C11 and C13. In the case of the third tested strain of *C. glabrata* (drug-resistant 2586), interactions with C9 and C15 were not observed, while with C11 and C13, MF = 2.

The results obtained during these experiments showed that AABs significantly increased the activity of antifungal drugs, and the strain of *C. glabrata* with the most robust resistance was 2853. 

#### 2.2.5. Influenced Combination of β-AE and Newly Synthesized AABs on Biofilm Eradication Multidrug-Resistant *C. glabrata*

The anti-biofilm (biofilm eradication) activity against the three strains of *C. glabrata* (reference strain ATCC 90030 and two multidrug-resistant strains 2586 and 2853) combinations of β-AE and the newly synthesized AABs was determined. The *C. glabrata* biofilm on the polystyrene surface (an abiotic surface) was treated with *β*-AE at the concentrations of 1 × MIC and 2 × MIC, with alkylamidobetaine derivatives (TMDAB) at the concentrations of 1, 2, and 3 µM, and with the combination of *β*-AE with the AABs (for each betaine derivative separately) in the system: 1 × MIC *β*-AE+1 µM, MIC *β*-AE+2 µM, and MIC *β*-AE+3 µM, and the same for 2 × MIC *β*-AE (forming six combinations of *β*-AE with TMDAB for each alkylamidobetaine altogether). The control was a biofilm not treated with any of the tested compounds. After 72 h, the ability to eradicate the *C. glabrata* biofilm of all the compounds and their combination based on the decreasing area of the biofilm and reduction of living cells was evaluated. It is worth emphasizing that all of the tested compounds, as well as their combination in a statistically significant manner, decreased both the percentage of the area of the biofilm and the number of living cells. In the case of β-AE (Figure 4), the concentration of 1 × MIC after 72 h decreased the area of the biofilm to 68.98% for the ATCC 90030 strain of *C. glabrata*, 59.3% for the 2586 drug-resistant strain, and 45.63% for the 2853 drug-resistant strain compared to the control (88.77%, 73.17%, and 84.52%, respectively). Decreasing biofilm viability was also noticed considering the % of live cells to 76.70% for the ATCC 90030 reference strain, 9.33% for the 2586 drug-resistant strain, and 92.13% for the 2853 multidrug-resistant strain of *C. glabrata* in comparison to the control (98.35%, 97.99%, and 97.46%, respectively). The highest susceptibility to β-AE is shown by the drug-resistant strain of *C. glabrata* 2586 concerning biofilm viability. The results for β-AE at a concentration of 2 × MIC are slightly better than for the concentrations of 1 × MIC (Figure 4).

The newly synthesized AABs (C9, C11, C13, and C15) at three concentrations (1, 2, and 3 µM) after 72 h of exposition decreased the % of the area of biofilm and its viability of *C. glabrata* of the drug-resistant strains. The range of the eradication ability of *C. glabrata* biofilms was generally from 15.95 to 65.45% for all AAB derivatives at all concentrations of biofilm area in comparison to the control (88.77%, 73.17%, and 84.52%, respectively). The tested AABs affected the biofilm surface of three strains of *C. glabrata* with different levels of effectiveness. For the ATCC 90030 reference strain, after 72 h of exposure to the AABs (in three different concentrations), the biofilm area was within the range of 15.95–57.97%, while for the drug-resistant strains 2586 and 2853, it was 39.84–65.45 and 23.01–50.26% of the biofilm area, respectively. Concerning the biofilm viability formatted by the three tested *C. glabrata* strains after 72 h of exposure to the tested AABs, the percentage of living cells in the biofilm was in the range of 1.96–83.56% for the ATCC 90030 reference strain, 2.33–12.91% for the 2586 drug-resistant strain, and 3.06–82.16% for the 2853 drug-resistant strain in comparison to the control. Moreover, for C9 and C11, the alkylamido derivatives’ decline in % of living cells was dose-dependent. Numerous details describing the influence of the tested compounds are presented in Figure 4.

Spectacular activity in the eradication of the *C. glabrata* biofilm in terms of the reduction in its surface area (expressed in %) was shown by the combinations of β-AE with the newly synthesized AABs.

After 72 h of treatment with the combination of β-AE (at concentrations of 1 × MIC and 2 × MIC) with the new AAB derivatives (at the concentrations of 1, 2 and 3 µM), the % of area of the biofilm was in the range of 1.74–27.74% for the ATCC 90030 reference strain of *C. glabrata*, while for the 2586 drug-resistant strain, it was 1.68–37.02, and for the 2853 drug-resistant strain, it was 0.89–8.74% in comparison to the control. Considering the viability of the *C. glabrata* biofilm (as a % of living cells), after 72 h of treatment with a mixture of β-AE with newly synthesized AABs, the percentage of living cells in the biofilm was in the range of 3.30–73.18% for the ATCC 90030 reference strain, while for the drug-resistant strains 2586 and 2853, they were 0.24–78.22 and 1.87–44.94%, respectively, in comparison to the control.

#### 2.2.6. Hemolysis Assays

Hemolysis assays are one of the tests performed in vitro to check the safety of the tested chemical compounds. It is necessary to evaluate the hemolytic potential of new substances with the possible application of indirect contact with human tissues.

β-AE was hemolytic at all concentrations tested, and all results agreed with the positive control (Figure 5). These results are unsurprising as saponins, to which β-AE belongs, exhibit hemolytic properties on red blood cells.

The newly synthesized AABs were hemolytic at 0.5, 1.0, 2.0, and 3.0 µM concentrations (Figure 6). Only at a concentration of 0.25 µM did the tested surfactants exert a hemolytic effect at an acceptable level (at the level of the negative control).

It should be emphasized that only the C9 alkylamidobetaine derivative was not hemolytic at all concentrations tested (Figure 6).

#### 2.2.7. Cell Proliferation Assays Balb/3T3 Mouse Embryonic Fibroblast (MTT Test)

The heat map in Figure 7 shows an analysis of the survival capacity of the fibroblast cells tested against the alkylamidobetaine–β-AE combinations tested.

Of the AABs that were taken for cytotoxicity testing, only C9 showed no cytotoxic effects (100% of surviving cells, corresponding to the green color on the heat map) on the fibroblast cells in the tested concentration range of 0.25 to 3 µM/mL (Figure 7). The others showed a cytotoxicity level of 85% for the concentrations of 1 µm/mL and 2 µM/mL for C11 and C13, respectively. In contrast, compound C15 decreased viability at the lowest concentrations tested, i.e., 0.25 and 0.5 µm/mL, by 15% and 28%, respectively (Figure 7). The β-AE showed strong toxic effects at all tested concentrations, i.e., 64–512 µM/mL (survival of the tested cell line at 83–85%). All tested alkylamidobetaine–β-AE combinations had toxic effects on the fibroblast cells, showing a 78–85% cytotoxicity, corresponding to the red and dark orange colors on the heat map.

The performed experiment showed that the tested compounds, in combination with the highly toxic β-AE, did not eliminate the harmful effects of this saponin on fibroblast cells.

## 3. Discussion

A vast increase in the frequency of fungal infections, especially among patients requiring long-term therapy and with an immune deficiency, is observed.

The WHO has recognized the problem, which has placed *C. glabrata* on the list of “priority pathogens”, strengthening the global response to fungal infections and resistance to antifungal drugs [12]. The *C. glabrata* strains very quickly attained drug resistance to azoles, which are drugs commonly used in the therapy of candidiasis due to the high bioavailability of this class of drug. Moreover, some *C. glabrata* strains are characterized by intrinsic resistance to this antifungal agent [12,36]. Even though azoles are the drugs of first choice in treating candidiasis, pathogens’ susceptibility has decreased dramatically in the last few decades, according to the development of various resistance mechanisms [36,37]. Therefore, the current stock of drugs used to combat fungal infections remains inadequate, and infections caused by *C. glabrata* strains resistant to a single drug and multiple drugs are becoming a common and difficult therapeutic challenge. 

One of the possible solutions to this urgent problem is to apply a combination of different drugs with azole therapy with great expectations for synergistic effects. Moreover, novel synergistic combinations may result in alternatives potentially lowering toxicity and costs and increasing effectiveness [38]. 

Searching for new efficient antifungal agents with minimal adverse effects and low toxicity often leads to abundant resources of natural substances, which are mainly present in plants [18,37,39] and bee products [40]. Compounds from natural products not only exhibit antifungal activity but also enhance the effects of synthetic drugs. 

The present study evaluated the antifungal activity of β-aescin (β-AE) alone and in combination with antifungal drugs against multidrug-resistant clinical strains of *C. glabrata*. Moreover, different combinations of β-AE with newly synthesized AABs as modern components eradicating the biofilm formatted by the mentioned *C. glabrata* strains were tested. β-AE belongs to the great family of natural surfactants called saponins, which are synthesized by different plants and involved in plant defenses through their antimicrobial potency [26,41]. Saponin extracted from various plants (roots, stem bark, or seeds) exerts pronounced activity against yeasts and filamentous fungi [42,43]. 

The results of our studies showed the activity of β-AE against the reference strain ATCC 90030 of *C. glabrata* (MIC_90_ = 128 µg; MFC = 256) and two drug-resistant strains: 2586 (MIC = MFC = 256 µg/mL) and 2853 (MIC = MFC = 128 µg/mL). The ratio of MIC = MFC indicates that this activity is fungicidal. Among all tested fungistatic agents (Table 2), only caspofungin and amphotericin showed lower MIC_90_ and MBC values for all tested strains. For 2853 drug-resistant strains of *C. glabrata*, lower MIC values were presented by fluconazole, ketoconazole, voriconazole, and posaconazole; however, very high MFC values for these azoles excluded their fungicidal activity.

The results obtained during our study referred to the evaluation of antifungal activity by Maione and colleagues [12], where β-AE was tested against three strains of *C. glabrata*: DSM 1226 and two clinical isolates (C18 and C27). The β-AEs showed fungicidal activity (the MFC/MIC ratio was less than four) for DSM 1226 with MIC = 80 µg/mL and MFC = 160 µg/mL, while for the clinical isolates C8 and C27, their values were MIC = 40 µg/mL and MFC = 100 µg/mL and MIC = 50 µg/mL and MFC = 100 µg/mL, respectively.

The in vitro antifungal activity of β-AE against 25 *C. glabrata* strains (including 24 clinical isolates and the reference strain ATCC 90030) was tested by Franiczek et al. [8]. The MIC values ranged from 8 to 32 µg/mL.

The activity of β-AE against multidrug-resistant strains of *C. glabrata* can be exploited in new treatment strategies, such as a combination approach, which can be an effective procedure for therapy-invasive candidosis [44].

A suitable combination of antimycotics can exhibit a more potent and broader antifungal effect and reduce the probability of incidence of resistance in fungi [18,45]. In the studies conducted by Maione and collaborators, the synergistic effect of the combination of β-AE with all three tested azole derivatives (fluconazole, itraconazole, and ketoconazole) against the biofilms of three tested *C. glabrata* strains (DSM 1226 and two clinical isolates, C18 and C19) was confirmed (FICI ≤ 0.5) [12]. In another study, the MIC values of nystatin (polyene antibiotics) were reduced by 2–16-fold and 2–4-fold in the presence of subinhibitory concentrations of β-AE crystalline and β-AE sodium, respectively [8]. The obtained results from these studies (conducted on the reference ATCC 90030 strain and 24 clinical isolates) may suggest the additive interaction between β-AE and nystatin [8]. 

Our results indicated a promising effect of the combination of β-AE with eight antifungal drugs and confirmed three types of interaction: synergism, addition, and indifference. The synergist effect only occurred when combining β-AE with fluconazole (FICI = 0.25) for the ATCC 90030 reference strain of *C. glabrata* (Figure 3). Moreover, the additive interaction between β-AE, voriconazole, posaconazole (FICI = 1.0), and caspofungin (FICI = 0.75) for ATCC 90030 was observed. For the drug-resistant strain 2586, an additive effect was noticed for a combination of β-AE with voriconazole, caspofungin (FICI = 1), and amphotericin B (FICI = 1.0). In the case of the drug-resistant 2853 strain, additive interactions were exerted by a combination of β-AE with fluconazole, ketoconazole, and caspofungin (FICI = 0.75), and posaconazole (FICI = 0.562). It is worth mentioning that the rest of the interactions had an indifferent character, and antagonism was not observed.

Our studies demonstrated the β-AE in vitro effect of *C. glabrata* biofilm eradication, indicated by decreasing biofilm area and reducing the percentage of living yeast cells in the biofilm. β-AE at a concentration of 1 × MIC decreased the biofilm area to 68.98% for the reference strain 90300 ATCC, 59.38 for the azole-resistant strain 2586, and 45.66% for the azole-resistant strain 2853 in comparison to the control (88.77%, 97.99%, and 84.52%, respectively). Decreasing *C. glabrata* biofilm viability was also noticed considering the % of live cells to 76.70% for the ATCC 90030 reference strain, 9.33% for 2586, and 92.13% for the 2853 multidrug-resistant strain of *C. glabrata* in comparison to the control (98.35%, 97.99%, and 97.46%, respectively). Increasing the concentration of β-AE to 2 MIC resulted in a further decrease in biofilm area (in %) and percentage of living cells in the biofilm (Figure 4).

The disruption of cell membrane integrity caused by β-AE affects other essential membrane functions, such as transport and signal transduction [46]. It also induces oxidative stress and free radicals, which amplify the damage in the cell [46]. The effect of the membrane integrity of *C. glabrata* cells by β-AE was confirmed through FACS analysis [12]. The FACS analysis demonstrated that β-AE affects the integrity of *C. glabrata* in a dose-dependent manner, indicating that β-AE at a concentration of 1 × MIC and 2 × MIC increased by about 7.4% and 22% of all volume, respectively, in comparison to the control [12]. 

Like polyene antibiotics (nystatin), β-AE reacts with the sterol molecules that make up the cell membrane, causing the formation of pores [8,12].

The anti-candidal activity of β-AE was confirmed by qRT-PCR analysis. The downregulation of *ERG11* (an ergosterol synthesis gene) and *ALS3* (a gene associated with adhesin production) demonstrates the great potential of this triterpene saponin as a natural antifungal agent to be further developed as part of novel anti-candidal strategies [12]. 

The results showed that new betaine derivatives with azoles, amphotericin B (antibiotic with a similar mechanism of action to azoles), and β-AE had a positive, from a therapeutic point of view, effect on *C. glabrata*, the ATCC reference strain ATCC 90030, and two drug-resistant clinical isolates: 2586 and 2853. However, we should mention that the drug-resistant 2852 strain has been labeled as having the lowest susceptibility.

The high efficacy of different types of surfactants against various microorganisms, including *Candida*, has led scientists to synthesize surface-active compounds, especially those that meet green chemistry standards.

C_n_TMDAB surfactants were rationally designed as a novel, multifunctional surface-active compounds with the intended characteristics in synthetic routes, purification steps, physicochemical characteristics, superior application performance, and optimal biological performance. The crucial point of interest constitutes the moderation of solid hydrogen bonding within the amide linkages in AABs—one of the most important groups of betaine-type zwitterionic surfactants with plenty of applications in different fields, from cosmetic and household cleaning formulations to enhanced oil recovery. Typically, AABs comprise a fatty acid chain attached to a betaine zwitterionic moiety by a secondary amide propyl-linking group (so-called AABs) and have to face numerous drawbacks, including limited aqueous solubility for long-chain derivatives, susceptibility to form gel-like structures or precipitate upon specific conditions, and insufficient ability to regulate formulation viscosity without the addition of sodium chloride, as well as their instability in hard water (i.e., vulnerability to separate into surfactant-rich and surfactant-poor phases in the presence of multivalent cations like Mg^2+^ and Ca^2+^). Considering the abovementioned drawbacks, our newly devised surfactants comprise a tertiary amide linking group (with the presence of a methyl moiety instead of hydrogen atom directly bonded to nitrogen). The general synthetic route meets the requirements of green and sustainable chemistry: primary raw materials (fatty acid derivatives) may be from natural sources, while the hydrophobic intermediates—*N*-[3-(dimethylamine)propyl]-*N*-methylalkylamides—may be obtained in solvent-less or mild conditions (e.g., transesterification, esterification, alkylation, etc.). At the same time, the crucial quaternization step was optimized to avoid unnecessary losses connected with a significant excess of quaternionic agents. Moreover, purification steps include commonly used crystallization procedures involving considerably smaller amounts of solvents than, e.g., preparative chromatography or liquid–liquid extraction. Some purification steps, especially the final crystallization step, may also be omitted for industrial-grade products. The used solvents or solvent mixtures may quickly return to the process after simple distillation. The synthetic route may only produce small amounts of side products, mostly non-toxic, like sodium chloride. It is worth noticing that the yield of the purified product is high (65-70%), so the formation of complex impurities (i.e., possessing similar surfactant molecules’ amphiphilic character) is limited.

It is well known that the solubility of betaine-type surfactants may be enhanced by NaCl additives, and the mentioned process provides significant viscosity improvements [47]. This phenomenon may be exploited for the viscosity regulation of household detergent formulations, shampoos, cosmetics, etc. On the other hand, the limited solubility of numerous amphoteric surfactants, especially AABs and their derivatives, may constitute a significant obstacle in their application, e.g., as mild disinfecting agents. Generally, long-chain betaine surfactants are characterized by high values of Krafft points, so their usage, especially in detergent compositions, is limited. In contrast to sulfobetaine and the sulfohydroxybetaines, alkylaminepropylbetaine-type surfactants are characterized by their low values of Krafft points (below 0 °C) even for very long hydrophobic chains (e.g., erucydyl derivatives). However, a certain amount of NaCl (ca. 3% wt in dry surfactants) is needed to assume the complete dissolution of the surfactant [47]. Our C_n_TMDAB surfactants are characterized by Krafft temperatures (see Table 1) below room temperature (25 °C) for all derivatives, with an exception for C_16_TMDAB (T_K_ = 36.4 °C). The results clearly show the high purity of the obtained zwitterions, especially the intense removal of NaCl traces; even a tiny amount of this salt would significantly depress the Krafft temperature. *N*-hexadecylamidepropylbetaine (with secondary amide linking group) [48] has a Krafft point equal to 16 °C, while its analog with a tertiary amide linker, C_16_TMDAB, was found to be characterized by T_K_ = 36.4 °C. This means that introducing a methyl group into the secondary amide linker, along with the change (reverse structure) of the linking moiety in the surfactant molecule, is needed to moderate hydrogen bonding in the dissolved surfactant.

Amphoteric and non-ionic surfactants are generally milder than cationic surfactants, reflected in their negligible eye, skin, and mucous membrane irritation and low antimicrobial activity [33]. This opinion was confirmed by our research, which showed that the newly synthesized AABs (CnTMDAB), when used alone, were practically inactive against the planktonic form of all *C. glabrata* strains tested (Table 1).

However, in the studies published by Liu [35], C12–C18 alkyl amido propyl dimethylamine betaine (AAPDB) complexed with chitosan acetate inhibited the growth of all organisms tested, including *C. albicans*. In addition, AAPDB was active against the tested yeast strain alone [35]. The most probable reason for the more potent antimicrobial activity exerted by the complex of chitosan acetate with AAPDB is the presence of the quaternary ammonium group with its ability to interact with the phospholipids of the cell membrane and disrupt them, as well as the ability to denature proteins (especially enzymes) [35]. In addition, introducing a C12–C18 alkyl chain will make the chitosan acetate/AAPDB complex more similar to the cell membrane and solubilize it, leading to the disintegration of the microorganism cell [35].

The mechanism of action of azole antifungals is to prevent the synthesis of ergosterol (a significant component of fungal plasma membranes) by inhibiting the CYP450-dependent enzyme lanosterol demethylase [49,50].

The mode of action of antifungal drugs with an azole ring in their structure may explain the increase in their activity against *C. glabrata* strains of antimycotics from 2-fold to 64-fold when combined with AABs (C9, C11, C13, and C15) at a constant concentration of 3 µM.

Recently, drug combinations have become an effective strategy to overcome the increasing resistance to candidal infections (especially *C. glabrata*). In the present work, we exploited the evaluated potentiality of AABs and β-AE combined with conventional azoles to affect membrane integrity, leading to the lysis of *C. glabrata* cells.

The antifungal properties of amphiphilic surfactants allow their potential use in many fields. According to their amphiphilic chemical character resulting from the presence of a hydrophilic head and an n-variable alkyl tail (hydrophobic), surface-active compounds can adsorb on different surfaces, covering them with layers. In addition, a surfactant-coated surface inhibits the ability of microorganisms to adsorb to it and form biofilm structures [51,52,53].

Several studies have suggested that hydrophobicity plays a crucial role in biofilm formation, so surfactants that can alter surface hydrophobicity may disrupt biofilm formation and cause its eradication [54].

The newly synthesized AABs (C9, C11, C13, and C15) used alone showed activity against biofilm formation by three tested strains of *C. glabrata*, particularly against the ATCC reference strain ATCC 90030 and the drug-resistant strain 2853. The activity of the tested compounds was dose-dependent for C9-C13 and better than β-AE used alone.

Spectacular results have been achieved by combining synthetic surfactants (alkylamidobetaine derivatives) with the natural surfactant β-AE.

Particularly noteworthy is the anti-biofilm activity of the combination of AE with C9 alkylamidobetaine, which, in optimal concentration configurations of both components, reduced the biofilm surface area of *C. glabrata* from 88.77% (control) to 1.37% for the reference strain ATCC 90030; from 73.17% (control) to 2.1% for the drug-resistant strain 2586; and, finally, from 84.52% (control) to 1.12% for the drug-resistant strain 2853.

The results for biofilm viability showed a similar trend. The percentage of live cells decreased after 72 h of treatment with the C9 AAB in combination with β-AE from 98.35% (control) to 1.64% for the reference strain *C. glabrata* ATCC 90030, from 97.99% (control) to 1.93% for the drug-resistant strain 2586, and from 97.46% (control) to 3.33% for the drug-resistant strain 2853.

It is therefore safe to say that the combination of natural (β-AE) and synthetic surfactants represents a new alternative method of controlling *C. glabrata* by eradicating the biofilm produced by this fungus. This combination will likely be equally effective as an anti-biofilm agent against other *Candida* pathogens.

Like all surfactants, newly synthesized AABs are characterized by their critical micelle concentration (CMC). The knowledge of the CMC of a particular surface-active compound is essential for many industries as it measures the surfactant’s efficiency, stability, and efficacy.

It can be concluded that as the alkyl chain length of AAB increases, the CMC value decreases. The exception to this rule is the C15 derivative, which has a higher CMC value than the C13 derivative. This overlaps to some extent with the results of the eradication activity of *C. glabrata* biofilms. When the AABs were tested separately (without the combination with β-AE), they showed a dose-dependent activity with increasing alkyl chain length (increasing hydrophobicity). This was true for the C9–C13 derivatives and broke down at C15.

In general, shorter alkyl chains of different surfactants are generally associated with weaker surface interactions and, consequently, weaker antifungal activity, with a specific gradation confirming the relationship between alkyl chain length, hydrophobicity, and observed activity [55].

In addition, other studies suggest a positive correlation between the length of the alkyl chain of surfactants and the ability to eradicate biofilms. High concentrations of surfactant compounds with fungicidal activity lead to the disintegration of the biofilm structure, which may be associated with micellization if the concentration exceeds the CMC [56].

The last stage of our studies was determining the side effects of the tested combination of β-AE and AABs against human cell lines. Our experiment used Balb/3T3 (ATCC CCL-163) mouse embryonic fibroblasts.

Of the AABs taken for cytotoxicity testing, only C9 showed no cytotoxic effects (100% of surviving cells). All tested AAB–β-AE combinations had a toxic impact on the fibroblast cells, showing a 78–85% cytotoxicity level, corresponding to the red and dark orange colors on the heat map. This experiment showed that the tested compounds, combined with highly toxic β-AE, did not eliminate the harmful effects of this saponin on fibroblast cells. Furthermore, hemolysis assays were performed for β-AE and all tested AABs. The results showed that β-AE was hemolytic at all concentrations tested, and all results agreed with the positive control (Figure 5). These results are unsurprising as saponins, to which β-AE belongs, exhibit hemolytic properties on red blood cells. Among the betaine derivatives, only C9 was not hemolytic at all concentrations tested. In conclusion, of all the compounds tested, only the C9 AAB was non-toxic at all doses tested.

## 4. Materials and Methods

In our research, we employed an array of materials and methods that will be presented in this section.

### 4.1. Compounds

List of compounds used in our research:

Newly synthesized AABs (TMDAB):**C_9_H_19_** molar mass: 328.49 (g/mol);**C_11_H_23_** molar mass: 356.54 (g/mol);**C_13_H_27_** molar mass: 384.60 (g/mol);**C_15_H_31_** molar mass: 412.65 (g/mol).

Patent no.: PL 237430 B1

Antifungal compounds:**β-aescin** (C_55_H_86_O_24_), CAS no.: 11072-93-8 (Sigma-Aldrich, Oakville, ON, Canada);**Fluconazole** (C_10_H_8_N_6_), CAS no.: 514222-44-7 (Glentham Life Sciences, Corsham, UK);**Itraconazole** (C_35_H_38_Cl_2_N_8_O_4_), CAS no.: 84625-61-6 (Glentham Life Sciences, Corsham, UK);**Ketoconazole** (C_26_H_28_Cl_2_N_4_O_4_), CAS no.: 65277-42-1 (Glentham Life Sciences, Corsham, UK);**Voriconazole** (C_16_H_14_F_3_N_5_O), CAS no.: 137234-62-9 (Glentham Life Sciences, Corsham, UK);**Posaconazole** (C_37_H_42_F_2_N_8_O_4_), CAS no.: 171228-49-2 (Glentham Life Sciences, Corsham, UK);**Caspofungin** (C_52_H_88_N_10_O_15_), CAS no.: 179463-17-3 (Glentham Life Sciences, Corsham, UK);**5-Fluorocytosine** (C_4_H_4_FN_3_O), CAS no.: 2022-85-7 (Glentham Life Sciences, Corsham, UK);**Amphotericin B** (C_47_H_73_NO_17_), CAS no.: 1397-89-3 (Glentham Life Sciences, Corsham, UK).

### 4.2. The Preparation and Chemical Characterization of AAB

#### 4.2.1. Materials

All the used reagents were of analytical or reagent grade and purchased from Sigma-Aldrich, with the exception of *N*, *N*, *N*’-trimethyl-1,3-propanediamine (Alfa Chemistry, NY, USA 99%). The other solvents (analytical grade) were obtained from Avantor Performance Materials (Gliwice, Poland). Triethylamine and tetrahydrofuran were dried and distilled over calcium hydride before use. The water that was used in all experiments was distilled twice and purified by a Millipore (Bedford, MA) Milli-Q purification system. The synthesis procedure of surfactants was carried out according to the procedures described in the Polish patent PL 237060.

#### 4.2.2. The synthesis of N-[3-(dimethylamine)propyl]-N-methylalkylamides

*N*, *N*, *N*’-trimethyl-1,3-propanediamine (10.0 g; 0.0861 mol) was dissolved in a 200 mL tetrahydrofuran–triethylamine (*v*:*v*, 1:1) mixture, after which the appropriate alkanoyl (decanoyl, dodecanoyl, tetradecanoyl, or hexadecanoyl) chloride (0.0861 mol) was added dropwise during intensive stirring. After that, the reaction mixture was stirred for 6 h at room temperature, followed by the filtration of triethylamine hydrochloride, the by-product. The filtrate was then evaporated and cautiously dried under reduced pressure to obtain N-[3-(dimethylamine)propyl]-N-methyoalkylamides as viscous liquids. The yield we obtained was 98–99.5%.

#### 4.2.3. The Synthesis of C_n_TMDAB

The appropriate *N*-[3-(dimethylamine)propyl]-*N*-methylalkylamide (0.04 mol), sodium chloroacetate (0.05 mol), and methanol (200 cm^3^) were placed in a pressure vessel and stirred at 90 °C for 40 h. After the completion of the reaction, the mixture was filtered, evaporated, and dispersed in anhydrous ethanol (200 cm^3^). The acquired mixtures were filtered and evaporated to dryness under reduced pressure. The products were recrystallized several times from methanol–ethyl acetate mixtures and dried over P_2_O_5_ under reduced pressure, yielding 55–70%.

#### 4.2.4. The Characterization of Betaine-Type Surfactants

CHNS elemental analysis was carried out by a Vario EL cube (Elementar, Germany) calibrated on acetanilide. Electrospray ionization mass spectroscopy (ESI-MS) (micrOTOF-Q instrument; Bruker Daltonics, Germany) allowed for the MS spectra to be obtained. The ESI-MS instrument was operated in the positive ion mode and calibrated with the Tunemix^TM^ mixture (Bruker Daltonics, Germany). The mass accuracy was better than 5 ppm. The obtained mass spectra were analyzed using DataAnalysis 3.4 software (Bruker Daltonics, Germany). ^1^H NMR spectra were recorded on a Bruker AMX-500 spectrometer, using CDCl_3_ as a solvent. ^1^H chemical shifts (in ppm) were calibrated to TMS as an internal reference. Melting points were determined using a Boetius melting point apparatus, while Krafft temperatures were determined according to the directions given in [47]. CMC is the minimum concentration above which micelles are formed in aqueous liquid. To obtain the CMC values, we used McGowan’s model. This method shows the relationship between partition coefficients and molecular volumes. Molecular characteristic volumes (V_x_) are found by the addition of the atomic characteristic volumes and the subtraction for the same factory (6.56 × 10^−6^ m^3^ mol^−1^) for every covalent bond.

### 4.3. Strains and Growth Condition

The *Candida glabrata* reference strain (ATCC 90030) was purchased from the American Type Culture Collection (LGC France SARL, Strasbourg, France), and 8 multidrug-resistant clinical strains (137, 769, 1467, 1941, 2342, 2586, 2783, and 2853) were obtained from the collection of the Department of Pharmaceutical Microbiology and Parasitology, Faculty of Pharmacy, Wroclaw Medical University. These strains were taken from patients of the Clinical University Hospital in Wroclaw. Strain 2586 was isolated from the vagina, while strain 2853 was obtained from bronchial secretions. These strains were characterized by a variable resistance phenotype within the group of azole drugs and 5-fluorocytosine (fluconazole, itraconazole, voriconazole, posaconazole, and 5-fluorocytosine were tested). Selected strains of *C. glabrata* with the highest ability to produce biofilm (2586 and 2853) were characterized by their resistance (R) to fluconazole, itraconazole, and posaconazole.

Yeast peptone glucose (YPG; 1% Difco Yeast extract, 1% Difco peptone, and 2% Difco glucose) was used to cultivate the strains. The obtained cultures were centrifuged, washed with PBS (pH 7.4), and suspended in fresh YPG so that a suitable optical density was achieved, according to the experimenter’s judgment.

### 4.4. Assessment of Biofilm Production Ability

The measurement was performed according to the modified method [51]. The biofilm on the microplate was stained using the LIVE/DEAD BacLight Bacterial Viability Kit (Thermo Fisher Scientific, Waltham, QC, Canada). For further information, see Section 4.12. below. This experiment was conducted with three independent repetitions.

Based on this test, a reference strain of *C. glabrata* (ATCC 90030) and 2 multidrug-resistant clinical strains (2586 and 2853) with the highest ability to produce biofilms were selected for further experiments.

### 4.5. Minimal Inhibitory and Fungicidal Concentrations

The values of the minimal inhibitory concentration (MIC) were obtained using the micro-dilution broth method, according to the European Committee on Antimicrobial Susceptibility Testing (EUCAST) (www.eucast.org; accessed on 1 November 2023), using the RPMI medium with MOPS (RPMI 1640 broth with 2% dextrose) and a spectrophotometric prominent inhibition of growth using Asys Hitachi 340, Driver version 4.02 (Biogenet, Poland) (optical density (OD) λ = 590 nm reduction of ≥90%) (MIC_90_), for the growth of fungi [57]. The sterile 96-well polystyrene microtitrate plates (Nunc, Denmark) were prepared by dispensing 100 µL of the appropriate dilution of the tested compounds in a broth medium per well by serial two-fold dilutions to obtain the final concentrations of the tested compounds that ranged from 4096 to 0.06 µg/mL. The inoculums that were prepared with fresh microbial cultures in sterile 0.85% NaCl to match the turbidity of the 0.5 McFarland standard were added to the wells to obtain a final density of 5 × 10^4^ CFU/mL for yeasts (CFUs: colony-forming units). An appropriate DMSO control (at a final concentration of 5%), a positive control (containing the inoculum without the tested compounds), and a negative control (containing the tested compounds without the inoculum) were included on each microplate.

The minimal fungicidal concentration (MFC) values were determined by inoculating 10 µL of the inoculum onto the YPG medium plate from the wells (MIC; 2 × MIC; and 4 × MIC) from a 96-well titration plate. The MFC was expressed as the concentration of the compounds that reduced the number of colony-forming units on the YPG medium (CFUs) by 99.9% after 24 h of incubation at 37 °C. Additionally, to confirm the MFC, the (3-(4,5-dimethylthiazol-2-yl)-2,5-diphenyltetrazolium bromide) MTT reagent was used, adding 10 µL to the wells at a final concentration of 5%. The lack of a color reaction (formazan formation) confirmed the result.

### 4.6. Fractional Inhibitory Concentration Index (FICI) of β-Aescin and Antifungal Compounds

A checkerboard method assessed the interactions of β-aescin (β-AE) with the selected antifungal compounds. Several antimycotics were investigated in these studies: fluconazole (FLU), itraconazole (ITR), ketoconazole (KET), voriconazole (VOR), posaconazole (POS), caspofungin (CAS), 5-fluorocytosine (5-FL), and amphotericin B (AMP). The β-aescin and antifungal agents listed above were diluted in the broth at the appropriate concentrations (based on their MIC values) ranging from 8 times higher than the MIC to 8 times lower than the MIC. These compounds were introduced horizontally (β-AE) and vertically (individual antimycotics) into the microplate. Then, after adding the *C. glabrata* strains’ (ATCC 90030, 2586, and 2853) inoculum to all of the wells, the plates were incubated as before [58]. After determining the MICs of β-AE alone and in combination, the fractional inhibitory concentrations (FICs) and FIC index (FICI, Σ FIC) were calculated as Σ FIC = FIC A + FIC B = (CA/MIC A) + (CB/MIC B), where MIC A and MIC B are the MICs of compounds A (β-AE) and B (studied antifungals) alone, respectively. In turn, CA is the MIC value of compound A in combination with B, and CB is the MIC value of compound B in combination with A. The interactions were interpreted based on the calculation of the FICI, in which: Σ FIC ≤ 0.5 synergism; 0.5 < Σ FIC ≤ 1 additivity; 0.5 < Σ FIC ≤ 1 neutral; and 1 < Σ FIC ≤ 4 antagonism interaction, respectively [18].

### 4.7. Interaction of the AABs with the Selected Antifungal Agents

A modified checkerboard method was used to assess the interactions of the newly synthesized AABs (C9, C11, C13, and C15) at the constant concentration of 3 µM with the selected antifungal compounds. Several antifungal compounds were investigated in these studies: β-aescin (β-AE), fluconazole (FLU), itraconazole (ITR), ketoconazole (KET), voriconazole (VOR), posaconazole (POS), caspofungin (CAS), 5-fluorocytosine (5-FL), and amphotericin B (AMP). The AABs and antifungal agents listed above were diluted in the broth (based on their MIC values) from 4096 µg/mL to 8x MIC depending on the mycostatic agent and proportionally going down 8 times. The AABs were introduced into the microplate wells (independent of compounds) and vertically (individual antimycotics). Then, after adding the *C. glabrata* strains’ (ATCC 90030, 2586, and 2853) inoculum to all of the wells, the plates were incubated for 24 h at 37 °C with shaking at 400 RPM. The MIC_90_ values were determined spectrophotometrically (λ = 590 nm) using Asys Hitachi 340, Driver version 4.02 (Biogenet, Jozefow, Poland). The multiplicity factor (MF) was calculated, determining the interactions between the AABs and antimycotics.

The formula used was follows: |MF| = MIC_90_ 1/MIC_90_ 2, where MIC 1 means the MIC value without additional compounds, and MIC 2 is influenced by alkylamidobetaine. This value is expressed in multiplicities.

### 4.8. Biofilm Eradication of Multidrug-Resistant C. glabrata Strains on Polystyrene Surfaces by β-Aescin and AAB Combination

*C. glabrata* biofilm eradication on polystyrene surfaces was assessed according to the following procedure: 1 mL of *C. glabrata* culture, 10^6^ CFU/mL in RMPI 1640 MOPS-buffered medium, was pipetted into the wells of a 48-well adherent plate. Cultures were incubated at 37 °C for 72 h with shaking (400 RPM). Subsequently, the wells were washed 3 times with sterile PBS (pH = 7.2); next, the chosen AABs (C9, C11, C13, and C15) were added to final concentrations ranging from 1 to 3 µM; for β-AE, it was added at a concentration of either 1 MIC or 2 MIC, and with the combination, it was incubated for 2 h at 37 °C with shaking (400 RPM). Non-treated cells were used as the control. This experiment was carried out in triplicate. The analysis of the biofilm eradication of *C. glabrata* strains on polystyrene surfaces was performed using fluorescence microscopy (FM). For further information, see Section 4.12. below.

### 4.9. Hemolysis Assay

This test was performed according to a modified method [59]. The hemolytic activity of the compounds was determined using sheep red blood cells. The blood was centrifuged (2000 RPM; 15 min) and rinsed 3 times with PBS. A total of 25 µL of erythrocytes for each sample were added to 500 µL of PBS to obtain the final concentration for the newly synthesized AABs: 0.25 µM, 0.5 µM; 1 µM, 2 µM; and 3 µM, and for β-AE, obtaining the following final concentrations: 100 µg/mL; 250 µg/mL; 500 µg/mL; 750 µg/mL; and 1000 µg/mL. The negative control (C−) consisted of erythrocytes suspended in PBS, and the positive control consisted of erythrocytes suspended in a 1% SDS solution. The samples were incubated for 2 h at 37 °C with shaking at 140 RPM. After this time, the blood cells were centrifuged (10 min; 2500 RPM), and 300 μL of the supernatant was transferred into a 96-well titration plate. Absorbance measurements were performed at a wavelength of λ = 540 nm. Measurements were performed in 10 repetitions. Finally, the values that were obtained from the samples treated with the test compound (OD test) were normalized relative to the positive (100% lysis; OD (C+)) and negative (untreated; OD (C−)) control samples to give the hemolysis ratio (HR) by using the following equation: HR (%) = OD test−OD (C−)/OD (C+) − OD (C−) × 100%.

### 4.10. Fibroblast Cell Culture

Balb/3T3 (ATCC CCL-163) mouse embryonic fibroblasts were cultured in DMEM medium supplemented with 10% fetal bovine serum, 1 mM L-glutamine, and a 1% penicillin/streptomycin solution and were used for our cytotoxicity studies. The cells were cultured at 37 °C and 5% CO_2_ until the cell density of 10^6^ mL^−1^ was obtained. Such cells were then seeded in 24-well tissue microplates (2 × 10^4^ cells per well) and allowed to adhere overnight under the abovementioned conditions. Cytotoxicity tests were carried out on the cells prepared in this way.

### 4.11. Cell Proliferation Assays (MTT Test)

The newly synthesized AABs (C9, C11, C13, and C15) were selected based on previous studies. They were analyzed at the concentrations of 0.25, 0.5, 1, 2, and 3 µM. The tested compounds were combined with β-AE at 64, 128, 256, and 512 µg/mL before being added to the cell line. The BALB-3T3 cells were treated with the prepared mixtures, dissolved in a complete cell culture medium, and then incubated for 24 h under the abovementioned conditions. Three independent replicates were performed for each combination of compounds.

The cytotoxic effect of the tested compounds in combination with β-AE was evaluated by measuring mitochondrial dehydrogenase activity using the MTT assay [60]. After a 24 h incubation period, the medium supplemented with the 1mM MTT solution was added to each well. The cultures were continued for another 4 h under the conditions enriched with 5% CO_2_. After this time, the supernatant was collected, and MTT formazan was extracted from the cells using DMSO. The reaction was then stopped with Sörensen buffer (0.1 mol/L glycine; 0.1 mol/L NaCl; and pH 10.5). The percentage of cytotoxicity was calculated by comparing the absorbance values (λ = 560 nm) of the test sample to the control sample, which was untreated fibroblast cells. In the test performed, the negative control (C−) (sample not treated; PBS added) and positive control (C+) were fibroblast cells treated with 1% SDS [60].

### 4.12. Fluorescence Microscopy (FM) and Computational Analysis of Pictures

The 72 h biofilm of *C. glabrata* strains on polystyrene surfaces was stained on 48-well adherent plates with 1 µL of propidium iodide (Ex λ = 543 nm/Em λ = 617 nm) and 1 µL of SYTO9 (Ex λ = 488 nm/Em λ = 503 nm) for 30 min using a LIVE/DEAD BacLight Bacterial Viability Kit (Thermo Fisher Scientific, Waltham, MA USA) [39]. Imaging was performed using a fluorescence microscope (FM) BX51 (Olympus, Tokyo, Japan) equipped with an Orca Flash 40 camera (Hamamatsu, Hamamatsu City, Japan) and 10× objective Plan APO. Scale bar = 100 µm. Detailed computer qualitative and quantitative analyses of the obtained pictures were performed by estimating the coverage area on the polystyrene plate (the percentage of the area occupied by the fungal biofilm). The cell viability analysis was performed by calculating the fluorescence intensity of the dyes (Syto9/PI) used. The acquired images were processed and analyzed using Fiji/ImageJ software ver. 1.53c (NIH, Bethesda, MD, USA). First, maximum intensity projections (MIPs) were obtained from stacks of images. The areas of the binary images were transferred into the original live and dead channel MIP images. The mean fluorescence intensities of all detected objects per field of view were calculated using ImageJ’s Analyze Threshold function (Software: ImageJ version 1.53 from Softonic).

### 4.13. Statistical Analysis

In this work, variance analysis was performed using Statistica 13 ver. 13.3.721.0 (StatSoft, Tulsa, OK, USA) (ANOVA analysis). Results for which *p* < 0.05 were treated as significant (indicating that the results are not identical at a 95% confidence level). All experiments were performed three times and at least in triplicate. The data were expressed as the mean ± standard deviation.

## 5. Conclusions

Forming biofilms by *C. glabrata* increases the probability of avoiding the response of the infected organism’s defense system. This leads to severe fungal infections, prolonging their duration and creating resistance to commonly used antifungal drugs.

The findings presented in this research showed the great potential for natural (β-aescin and synthetic surfactants as antifungal agents used alone or with conventional antimycotics (azoles and antifungal antibiotics). Moreover, a combination of β-aescin with newly synthesized (following green chemistry principles) AABs offers excellent potential to eradicate the biofilms of *C. glabrata*. In conclusion, synergistic interactions between β-AE, AAB, and some well-known but not always efficient antimycotics may contribute to developing new antifungal strategies and formulations.


**The highlights of our research are as follows:**
AAB compounds have the potential to be used as adjuvant compounds due to their demonstrated synergy and additive effects with selected antifungal drugs.The combination of β-AE with AABs is a highly effective mixture for eradicating *C. glabrata* biofilms.Due to its low cytotoxicity and hemolytic potential, the AAB C9 may be used in practice after further clinical trials.


## 6. Patents

Patent no.: PL 237430 B1.

## Figures and Tables

**Figure 1 ijms-25-02541-f001:**
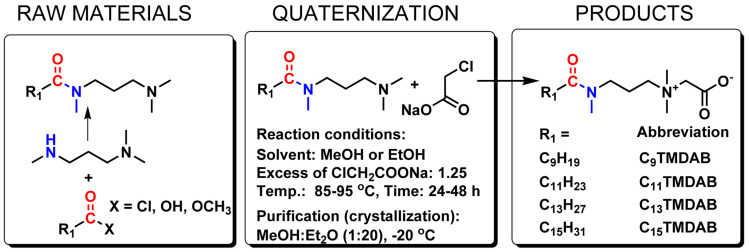
Chemical structures, abbreviations, and synthetic routes for novel [(3-alkanoyilomethyoamine)propyl] dimethylammonium acetates (C_n_TMDAB).

**Figure 2 ijms-25-02541-f002:**
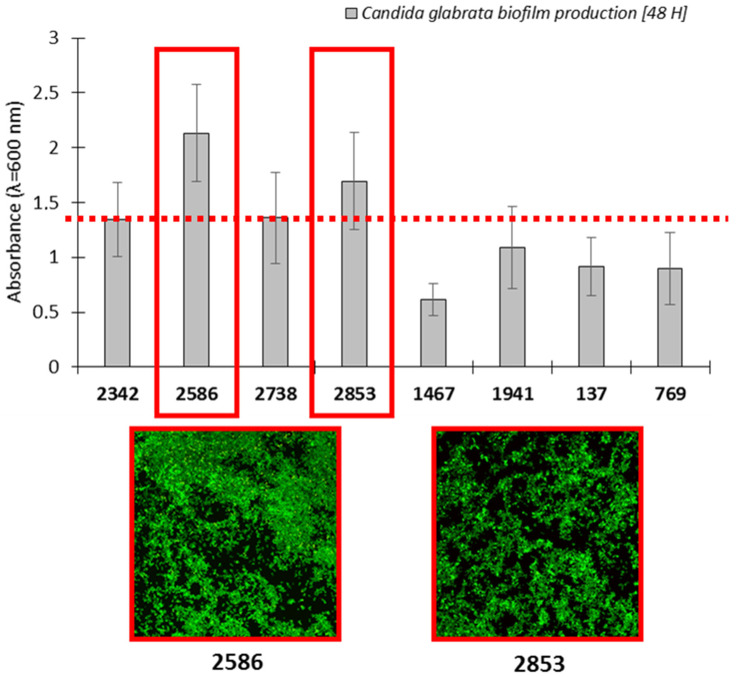
This graph shows the level of biofilm production by the tested drug-resistant clinical *C. glabrata* strains and representative pictures of biofilm formation detected through fluorescence microscopy. Scale bar = 100 µm.

**Figure 3 ijms-25-02541-f003:**
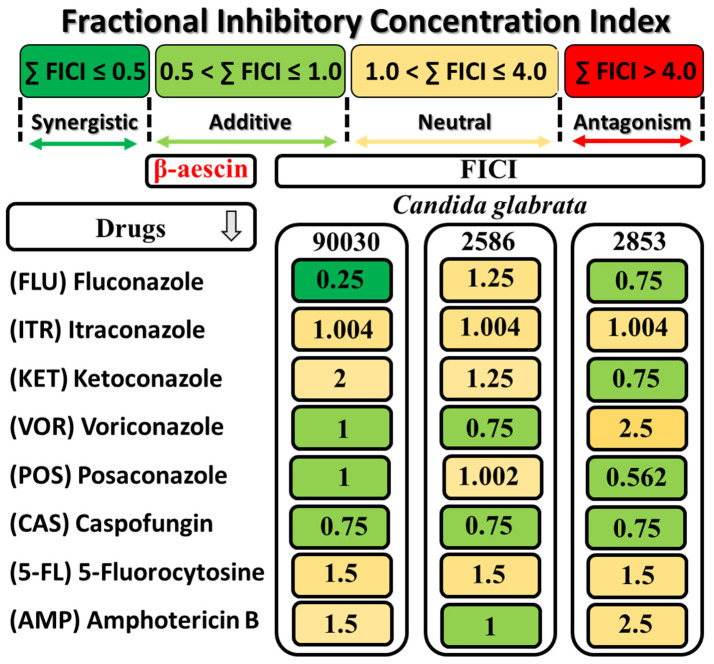
A graphical representation of the results of the interactions between β-AE and antifungal drugs against *C. glabrata* strains. The fractional inhibitory concentration index (FICI) calculation interpreted the interactions.

**Figure 4 ijms-25-02541-f004:**
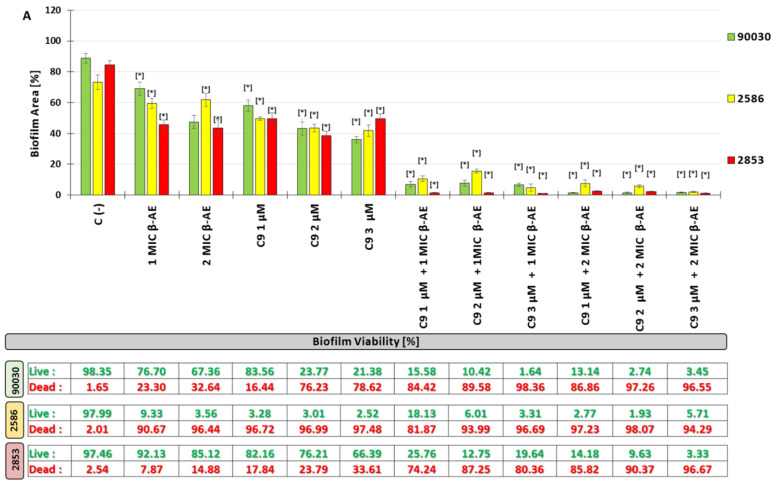
Influenced combination of β-AE and the newly synthesized AABs (**A**) C9; (**B**) C11; (**C**) C13; and (**D**) C15 on biofilm eradication (72 h) against drug-resistant *C. glabrata*; mean ± SD, n = 3; * statistically different from the control, *p* < 0.05. The graph shows the actual biofilm area on the analyzed surface. Below the graphs, the biofilm viability tables (ratio of living to dead structures—Syto9/PI staining) are presented based on the analyzed surfaces.

**Figure 5 ijms-25-02541-f005:**
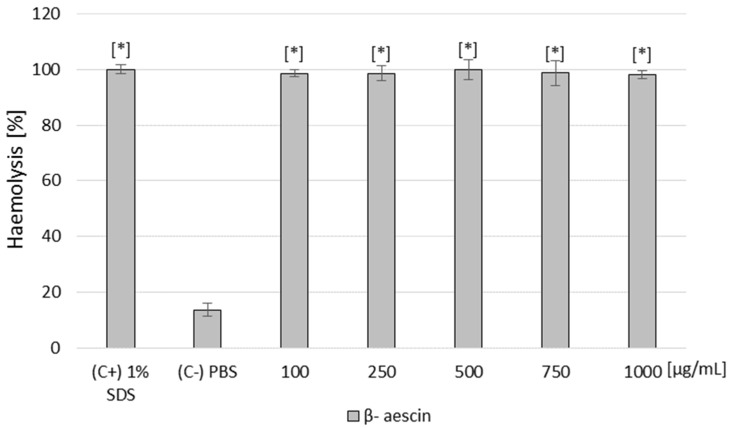
Effect of β-AE on the hemolysis of sheep blood cells (in concentrations from 100 up to 1000 µg/mL). Legend: (C+): positive control—1% SDS; (C−): negative control—not treated; * statistically different from the control (−), *p* < 0.05.

**Figure 6 ijms-25-02541-f006:**
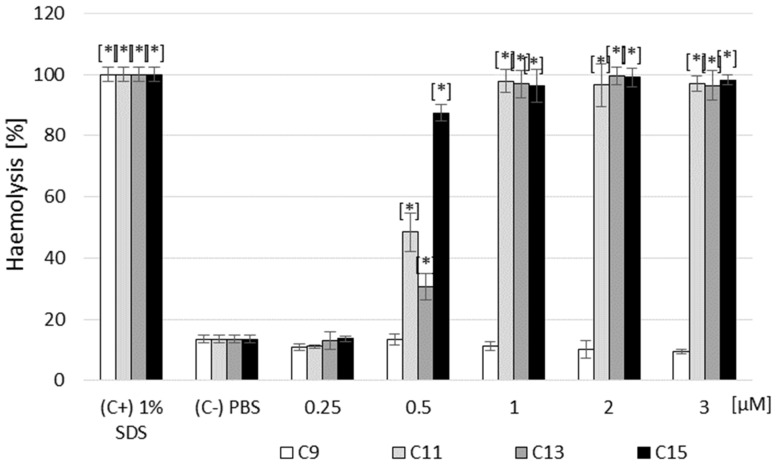
Effect of the newly synthesized AABs (C9; C11; C13; and C15) on the hemolysis of sheep blood cells (in concentrations from 0.25 up to 3 µM). Legend: (C+): positive control—1% SDS; (C−): negative control—not treated; * statistically different from the control (−), *p* < 0.05.

**Figure 7 ijms-25-02541-f007:**
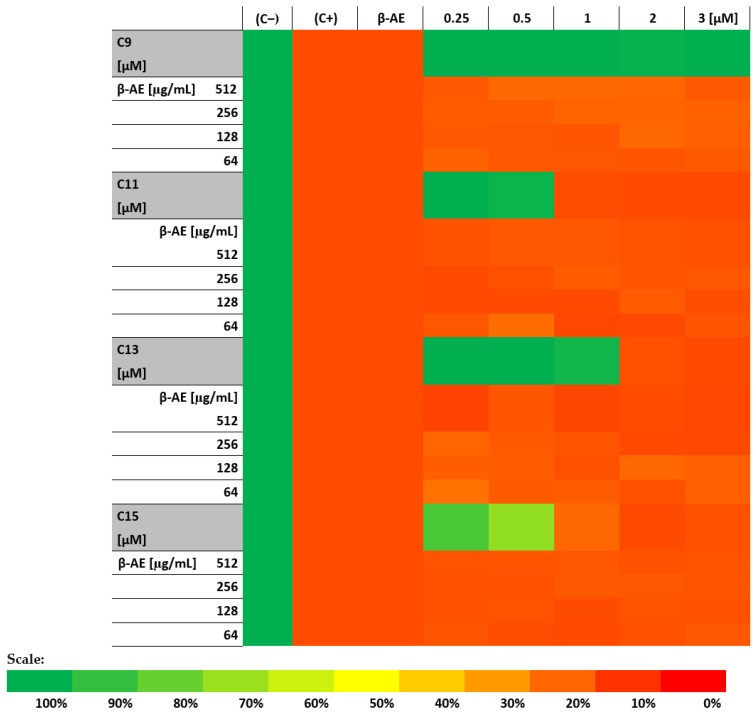
Distribution of fibroblast cell survival after treatment with the tested combinations of β-AE and newly synthesized AABs (C9; C11; C13; and C15). Legend: β-AE (β-aescin); (C+): positive control—2% SDS; (C−) negative control—not treated.

**Table 1 ijms-25-02541-t001:** Properties and spectroscopic data of the studied [(3-alkanoyilomethyoamine)propyl] dimethylammonium acetates (C_n_TMDAB).

Chemical structure	R	Molecular Weight (g/mol)	^1^H NMR, CDCl_3_	ESI-MS (MNa^+^)	Elemental Analyses (Calculated)			Melting Point (°C)	T_K_ (°C)	CMC (mol/dm^3^)	Abbreviation
			δ (ppm)		%C	%H	%N				
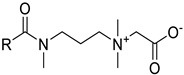	C_9_H_19_	328.49	0.84–0.91 [t, 3H, -COCH_2_CH_2_(CH_2_)_k_CH_3_]; 1.22–1.34 [m, 2kH, -COCH_2_CH_2_(CH_2_)_k_CH_3_]; 1.57–1.60 [m, 2H, -COCH_2_CH_2_(CH_2_)_k_CH_3_]; 1.99 [m, 2H, -N^+^CH_2_CH_2_CH_2_N-]; 2.28–2.31 [t, 2H, -COCH_2_CH_2_(CH_2_)_k_CH_3_]; 3.02–3.05 [s, 3H, -N(CH_3_)-]; 3.34–3.45 [s, 6H; -N^+^(CH_3_)_2_-]; 3.63–3.67 [m, 4H, -N^+^CH_2_CH_2_CH_2_N-]; 3.99–4.03 [s, 2H, -N^+^CH_2_COO^−^].	351.5	65.79 (65.81)	11.10 (11.07)	8.52 (8.53)	-	<0	3.0 × 10^−2^	**C9**
	C_11_H_23_	356.54		379.5	67.31 (67.37)	11.30 (11.33)	7.91 (7.86)	45–47	~5	8.7 × 10^−3^	**C11**
	C_13_H_27_	384.60		407.6	68.65 (68.70)	11.52 (11.55)	7.26 (7.29)	58–59	20.5	2.5 × 10^−3^	**C13**
	C_15_H_31_	412.65		435.6	69.84 (69.85)	11.77 (11.75)	6.78 (6.79)	70–71	36.4	7.4 × 10^−3^	**C15**

**Table 2 ijms-25-02541-t002:** Minimum inhibitory concentrations, in µM/mL (MIC_90_), of the AABs with a general structure of CnTMDAB (C9; C11; C13; and C15) (µM) against *C. glabrata* strains.

R	M_w_ (g/mol)	Abbreviation	ATCC 90030	2586	2853
			MIC_90_	MIC_90_	MIC_90_
C_9_H_19_	328.49	**C9**	3	3	1
C_11_H_23_	356.54	**C11**	>3	>3	>3
C_13_H_27_	384.60	**C13**	>3	>3	>3
C_15_H_31_	412.65	**C15**	>3	>3	>3

**Table 3 ijms-25-02541-t003:** Minimum inhibitory concentrations (**MIC_90_**) and minimum fungicidal concentrations (**MFC**) of compounds (µg/mL) against *C. glabrata* strains.

Abb.	Compound	ATCC 90030		2586		2853	
		MIC_90_	MFC	MIC_90_	MFC	MIC_90_	MFC
**β-AE**	β-aescin	128	256	256	256	128	128
**FLU**	Fluconazole	2048	2048	2048	2048	32	2048
**ITR**	Itraconazole	4096	4096	4096	4096	4096	4096
**KET**	Ketoconazole	256	1024	512	1024	4	1024
**VOR**	Voriconazole	512	1024	512	1024	2	512
**POS**	Posaconazole	256	1024	512	1024	1	256
**CAS**	Caspofungin	0.06	0.06	0.03	0.06	0.03	0.06
**5-FL**	5-Fluorocytosine	512	1024	512	1024	256	512
**AMP**	Amphotericin B	0.25	0.5	0.25	1	0.25	0.5

**Table 4 ijms-25-02541-t004:** Interaction of AABs with selected antifungal agents. Antimycotics combined with AABs (C9, C11, C13, and C15) at a constant concentration of 3 µM against *C. glabrata* strains.

Abb.	Compound	ATCC 90030				2586				2853			
		C9	C11	C13	C15	C9	C11	C13	C15	C9	C11	C13	C15
**β-AE**	β-aescin	128 **(1)**	128 **(1)**	128 **(1)**	128 **(1)**	128 **(2)**	128 **(2)**	128 **(2)**	128 **(2)**	128 **(1)**	128 **(1)**	128 **(1)**	128 **(1)**
**FLU**	Fluconazole	128 **(16)**	32 **(64)**	32 **(64)**	32 **(64)**	256 **(8)**	32 **(64)**	32 **(64)**	32 **(64)**	32 **(1)**	16 **(2)**	16 **(2)**	2 **(16)**
**ITR**	Itraconazole	4096 **(1)**	4096 **(1)**	4096 **(1)**	4096 **(1)**	4096 **(1)**	4096 **(1)**	4096 **(1)**	4096 **(1)**	4096 **(1)**	4096 **(1)**	4096 **(1)**	4096 **(1)**
**KET**	Ketoconazole	16 **(16)**	16 **(16)**	16 **(16)**	4 **(64)**	64 **(8)**	32 **(16)**	16 **(32)**	8 **(64)**	4 **(1)**	4 **(1)**	2 **(2)**	1 **(4)**
**VOR**	Voriconazole	8 **(64)**	8 **(64)**	8 **(64)**	8 **(64)**	64 **(8)**	8 **(64)**	8 **(64)**	8 **(64)**	2 **(1)**	1 **(2)**	1 **(2)**	256 **(128)**
**POS**	Posaconazole	512 **(2)**	32 **(8)**	32 **(8)**	512 **(2)**	512 **(1)**	256 **(2)**	256 **(2)**	512 **(1)**	8 **(8)**	4 **(4)**	4 **(4)**	256 **(256)**
**CAS**	Caspofungin	0.06 **(1)**	0.06 **(1)**	0.06 **(1)**	0.03 **(1)**	0.03 **(1)**	0.03 **(1)**	0.03 **(1)**	0.03 **(1)**	0.03 **(1)**	0.03 **(1)**	0.03 **(1)**	0.03 **(1)**
**5-FL**	5-Fluorocytosine	512 **(1)**	512 **(1)**	512 **(1)**	512 **(1)**	512 **(1)**	512 **(1)**	512 **(1)**	512 **(1)**	256 **(1)**	256 **(1)**	256 **(1)**	256 **(1)**
**AMP**	Amphotericin B	0.25 **(1)**	0.12 **(2)**	0.12 **(2)**	0.06 **(4)**	0.12 **(2)**	0.12 **(2)**	0.12 **(2)**	0.06 **(4)**	0.25 **(1)**	0.12 **(2)**	0.12 **(2)**	0.06 **(4)**

Multiplicity factor (bold values in brackets). Legend: green—increasing activity; black—neutrality; and red—decreased activity.

## Data Availability

Data are contained within the article.

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
