# Peer review of "A Combination of β-Aescin and Newly Synthesized Alkylamidobetaines as Modern Components Eradicating the Biofilms of Multidrug-Resistant Clinical Strains of Candida glabrata"

_ijms, 2024, doi:10.3390/ijms25052541_

Round 1

Reviewer 1 Report

Comments and Suggestions for Authors

In this paper, the authors present the results of the study of the possibility of combining the natural saponin β-aescin with recently synthesized alkyladobetaines with different chain lengths of alkyl substituents and (antifungal drugs. A large number of studies have been carried out by the authors. In addition to testing the antibiofilm activity of the obtained drug combinations, the cytotoxicity and biosafety were also evaluated using fibroblasts and erythrocytes.  The relevance of this work is due to the practical value of the results presented in it, due to the spread of fungal strains resistant to standard drugs. The results presented in the article are neatly organized. The study expands the knowledge about practical possibilities of using antibiofilm compounds in combination therapy against C. glabrata. There are a number of comments and suggestions to improve the article.

1. There is no reference in the list of references 47 

2. It is required to bring to uniformity the design of the list of references. Almost every reference has errors in design.

3. Given that the authors have chosen Candida glabrata culture as the objects of research in this paper, the introduction should not start with a discussion of the problem of antibiotic resistance, but with a discussion of the problem of mycosis spread.

4. the authors should explain the relevance of the search for new antibiotic drugs for the treatment of mycoses in the practical use of a number of antibiotic (mucolytic) drugs. 

5. The authors should remove the yellow highlighting in Table 3.

6. Discussion The authors should correct this section and remove repetitive sentences from the text, in particular on the resistance of C. glabrata to azoles. 

7. The authors should add a general conclusion on the results obtained.

Author Response

File in attachment

Reviewer 2 Report

Comments and Suggestions for Authors

The paper is very interesting, containing a wide variety of methods regarding the study of yeast cell biology and its susceptibility to new compounds.
However, the authors are not without errors, mainly editorial. The whole text is rather difficult to read, there are many punctuation errors, additional unnecessary characters, some entries are ambiguous and some phrases are underdeveloped. Here are some of them:
Line 162: "Among the 8 multidrug-resistant strains, C. glabrata 2586 and 2853 " which eight strains are being referred to? what is their multidrug resistance? what is their susceptibility phenotype?

Lines 162-164: The names of clinical and reference strains should be unambiguous. The notation of the reference strain should be as follows C. glabrata ATCC 90030. The abbreviation ATCC should be clarified. What is the origin of the clinical strains?

Table 1: Units of MIC concentrations are missing from the table, analogous to the second column, as well as "Abb." shortcut meaning.

Lines 193-207: Please harmonise the units for MIC and MFC concentrations - the authors give μM/mL and μg/mL once. The same applies to the notation of the reference strain - in the following paragraphs it is already correct "C. glabrata ATCC 90030", instead of "the 90030 ATTC strain" or "C. glabrata 90030 ATCC" or "reference strain 90030 ATCC"

Line 204: The phrase "........which represents 985, 47 μg/mL and for 2853 clinical isolate MIC=1 204 μM/mL=328. 49 μg/mL........" is unclear

Line 277: does this 'concentrations 1 MIC and 2 MIC' mean concentrations 1xMIC and 2xMIC?

Line 734: "99.9% after 24 h of incubation at 37 °C25" number 25 is unnecessary, simmilarly, line 736 "the CLSI26"

Line 732: "MFC was expressed as the concentration of the compounds that reduced the number of colony-forming units on YPG medium (CFU) by 99.9% after 24 h of incubation at 37 °C25", also the sentence is incomprehensible. The authors claim that they performed the MFC determination in accordance with the CLSI recommendation and yet this sentence contradicts this."

There is too much confusion in the numbering of references.

In my opinion, the work as it stands cannot be accepted for publication because it is underdeveloped. It is obvious that it has been written by several members of the team with different writing styles.

Comments on the Quality of English Language

The language of the work needs to be more refined.

Author Response

File in attachment

Reviewer 3 Report

Comments and Suggestions for Authors

See Report

Author Response

File in attachment

Round 2

Reviewer 3 Report

Comments and Suggestions for Authors

Despite my recommendation to shorten the manuscript, particularly in the Introduction, Results, and Discussion sections, no significant reductions were made.

Introduction:

The Introduction was not reduced, and paragraphs suggested to be moved to Discussions or Conclusions remained intact.

Results:

The Results section was not shortened or reorganized, despite being overly long and repetitive. Some paragraphs resemble discussion rather than results, some paragraphs repeat or complete the Material and Methods and many reiterate information already presented in the Tables.

Discussion:

Similarly, the Discussion section was not shortened as requested, despite the authors' claim that they had addressed this. The discussion remains excessively lengthy and could benefit from substantial reduction.

Additionally, I requested more discussion on toxicity, as it appears to be a significant concern given the strong toxic effects observed at all tested concentrations. However, no changes were made in this regard, despite the authors' assertion that they had added these discussions.

Furthermore, the limitations of the study were not addressed, although I emphasized their importance.

Materials and Methods:

There was minimal improvement in the Materials and Methods section, contrary to the authors' claim that a significant part of it had been rewritten.

Overall, the manuscript has not adequately addressed the recommendations for reduction and clarification, and significant revisions are still necessary.

Author Response

Dear reviewer, we have added new suggested corrections to the new version of the manuscript. Thank you very much for your commitment.

First, the manuscript is too long and should be shortened in the sections: Introduction, Results, Discussion.

Title

Abbreviations should not be used: Candida glabrata instead of C. glabrata.

DONE

Abstract

Line 21-22: “The first concerns determining biological activity against” - it is unclear whose activity is determined

Corrected

Line 22: Candida glabrata instead of C. glabrata - The names of the bacteria must first be written in full and then only abbreviated

Corrected

Introduction

It can be shortened.

Lines deleted:

Line 76-77

Line 87-89

Line 122-128

Line 122-127: it should not be in the Introduction, but in the Discussions

Deleted

Table 1: it should not be in the Introduction, but in the Results

It is moved.

Line 129-131: it should not be in the Introduction, these are conclusions

At the end of the Introduction, a paragraph about the purpose of the study must be inserted.

The purpose of the study has been inserted.

Results

They are too long; they need to be shortened and reorganized to be clear and concise.

Some paragraphs repeat or complete the Material and Methods: e.g. Line 157-177, Line 210-216, Line 274-284.

Line 155-177 represent clear chemical  properties of synthetized AAB (Results).

Line 210-216 were deleted.

Line 274-284 describing results of antifungal activity.

Line 334-336: these are discussion, not results.

Line 334-336 describing results of antibiofilm activity.

Many paragraphs repeat the results from the Tables.

The numbering of the tables is wrong: Table 1 appears on page 4 and page 5.

In Table 1, page 5, the structure is not required, only the results from the MIC.

Corrected.

Discussion

They are much too long and should be shortened.

Lines deleted:

Lines 213-215

Lines 297-298

Lines 393-395

Lines 445-447

Lines 477-480

Lines 490-491

Lines 605- 606

Lines 628-631

Round 3

Reviewer 3 Report

Comments and Suggestions for Authors

Although the authors did not follow all my suggestions, the manuscript has been improved and I think it can be published if the editors agree.